# Molecular Features and Clinical Management of Hereditary Pancreatic Cancer Syndromes and Familial Pancreatic Cancer

**DOI:** 10.3390/ijms23031205

**Published:** 2022-01-21

**Authors:** Akiyoshi Kasuga, Takeshi Okamoto, Shohei Udagawa, Chinatsu Mori, Takafumi Mie, Takaaki Furukawa, Yuto Yamada, Tsuyoshi Takeda, Masato Matsuyama, Takashi Sasaki, Masato Ozaka, Arisa Ueki, Naoki Sasahira

**Affiliations:** 1Department of Hepato-Biliary-Pancreatic Medicine, Cancer Institute Hospital of Japanese Foundation for Cancer Research, Tokyo 135-8550, Japan; takeshi.okamoto@jfcr.or.jp (T.O.); chinatsu.mori@jfcr.or.jp (C.M.); takafumi.mie@jfcr.or.jp (T.M.); takaaki.furukawa@jfcr.or.jp (T.F.); yuto.yamada@jfcr.or.jp (Y.Y.); tsuyoshi.takeda@jfcr.or.jp (T.T.); masato.matsuyama@jfcr.or.jp (M.M.); takashi.sasaki@jfcr.or.jp (T.S.); masato.ozaka@jfcr.or.jp (M.O.); naoki.sasahira@jfcr.or.jp (N.S.); 2Department of Medical Oncology, Cancer Institute Hospital of Japanese Foundation for Cancer Research, Tokyo 135-8550, Japan; shohei.udagawa@jfcr.or.jp; 3Department of Clinical Genetics, Cancer Institute Hospital of Japanese Foundation for Cancer Research, Tokyo 135-8550, Japan; arisa.ueki@jfcr.or.jp

**Keywords:** hereditary pancreatic cancer syndrome, familial pancreatic cancer, germline mutation, loss of heterozygosity, surveillance, PARP inhibitor

## Abstract

Hereditary pancreatic cancers are caused by several inherited genes. Familial pancreatic cancer is defined as pancreatic cancer arising in a patient with at least two first-degree relatives with pancreatic cancer in the absence of an identified genetic cause. Hereditary pancreatic cancer syndromes and familial pancreatic cancers account for about 10% of pancreatic cancer cases. Germline mutations in *BRCA1, BRCA2, ATM, PALB2, CDKN2A, STK11,* and *TP53* and mismatch repair genes (*MLH1, MSH2, MSH6, PMS2,* and *EPCAM*) are among the well-known inherited susceptibility genes. Currently available targeted medications include poly (ADP-ribose) polymerase inhibitors (PARP) for cases with mutant *BRCA* and immune checkpoint inhibitors for cases with mismatch repair deficiency. Loss of heterozygosity of hereditary pancreatic cancer susceptibility genes such as *BRCA1/2* plays a key role in carcinogenesis and sensitivity to PARP inhibitors. Signature 3 identified by whole genome sequencing is also associated with homologous recombination deficiency and sensitivity to targeted therapies. In this review, we summarize molecular features and treatments of hereditary pancreatic cancer syndromes and surveillance procedures for unaffected high-risk cases. We also review transgenic murine models to gain a better understanding of carcinogenesis in hereditary pancreatic cancer.

## 1. Introduction

Pancreatic cancer is one of the leading causes of cancer-related mortality worldwide. The 5-year survival rate in pancreatic cancer remains below 10% [1]. Despite advances in new systemic treatments and the increased availability of active agents, median overall survival (OS) in metastatic pancreatic cancer is less than one year [2,3,4,5].

Multiple combinations of somatic gene mutations have been identified as causes of pancreatic cancer. Mutational activation of oncogenic *KRAS* and inactivation of tumor suppressor genes such as *TP53*, *CDKN2A,* and *SMAD4* are the four major driver genes in pancreatic cancer [6]. Several genes responsible for familial occurrences of pancreatic cancer have also been identified [7]. Up to 10% of pancreatic cancers have been reported to be related to inherited genes [8,9,10,11,12].

There are two broad risk categories in inherited pancreatic cancer. Hereditary pancreatic cancer syndromes are defined as inherited genetic predisposition syndromes related to pancreatic cancer. Familial pancreatic cancer (FPC) is defined as pancreatic cancer arising in a patient with at least two first-degree relatives (FDRs) with pancreatic cancer in the absence of known pancreatic cancer susceptibility genes. Most hereditary pancreatic cancers with genetic factors have autosomal dominant inheritance, presenting a 50% probability that the pathogenic variant will be passed on to the next generation. After germline mutations in tumor suppressive genes are inherited, inactivation of both alleles in combination with mutations of other driver genes is necessary for carcinogenesis. As a general rule, the second allele is inactivated by somatic mutation, loss of heterozygosity, or epigenetic silencing by promoter hypermethylation. In a subset of germline mutation carriers, pancreatic cancers may arise independent of the germline mutation [13,14,15]. Although *KRAS* mutation is frequent in pancreatic cancer, biallelic inactivation of Brca2 promoted chromosomal instability and apoptosis and inhibited carcinogenesis in the presence of *Kras* activation in murine hereditary pancreatic cancer models with intact *Trp53* [13,14].

Several groups have published surveillance guidelines for unaffected relatives of hereditary pancreatic cancer syndromes and families with FPC [12,16]. The main purpose of screening for high-risk individuals is the detection of precursor lesions or early pancreatic cancer that can be resected curatively. Advances are also being observed in precision medicine for hereditary pancreatic cancer syndromes. Clinical development of poly (ADP-ribose) polymerase (PARP) inhibitors for cases with homologous recombination repair (HRR) deficiency and immune therapies for cases with mismatch repair deficiency appear promising.

This review summarizes the current literature on hereditary pancreatic cancer and FPC and clinical management. We also focus on murine models of hereditary pancreatic cancer to gain a better understanding of carcinogenesis in hereditary pancreatic cancer. 

## 2. Hereditary Pancreatic Cancer Syndromes

A wide variety of hereditary cancer-related genes that can cause cancer in multiple organs have been identified. Hereditary pancreatic cancer overlaps with hereditary cancer syndromes. Hereditary pancreatic cancer syndromes with specific germline mutations are summarized in Table 1.

### 2.1. Hereditary Breast and Ovarian Cancer Syndrome (HBOC)

Hereditary breast and ovarian cancer (HBOC) is characterized by the presence of pathogenic variants identified in breast cancer associated (*BRCA*) 1 and 2. *BRCA1* and *BRCA2* are canonical tumor suppressor genes involved in DNA damage repair through the HRR pathway. The most common cause of hereditary pancreatic cancer is a germline mutation in BRCA2.

Pathogenic BRCA1/2 variants increase the risk of breast, ovarian, pancreatic, prostate, and other cancers. BRCA1/2 mutant carriers have a relatively low risk of pancreatic cancer compared to breast and ovarian cancer. In FPC registry studies, *BRCA2* mutant carriers have been identified in 5–17% of patients with FPC and *BRCA1* mutations were not highly prevalent [17,18,19,20]. Overall risk for breast cancer and ovarian cancer is 55–70% and 40–45% for *BRCA1* mutant carriers, respectively, and 45–70% and 15–20% for *BRCA2* mutant carriers, respectively [21,22,23,24]. The relative risk for breast cancer and ovarian cancer is 5.9 (95% confidence interval (CI): 5.3–6.7) and 11.8 (95% CI: 10.0–14.0) for *BRCA1* mutant carriers, respectively, and 3.3 (95% CI: 3.0–3.7) and 5.3 (95% CI: 4.4–6.3) for *BRCA2* mutant carriers, respectively [25]. The lifetime risk of pancreatic cancer is about 1% for *BRCA1* mutant carriers and 4.9% for *BRCA2* mutant carriers [26,27]. The relative risk for pancreatic cancer in *BRCA1* and *BRCA2* mutant carriers is 2.3 and 3.5–10, respectively [26,28,29]. Compared to breast cancer and ovarian cancer, *BRCA1/2* mutations alone do not pose a significant risk of pancreatic cancer.

### 2.2. PALB2

*PALB2* is the partner and localizer of *BRCA2*. The *PALB2* protein binds with the *BRCA2* protein and stabilizes it in the nucleus. Germline mutations of *PALB2* increase the risk of pancreatic cancer; although, the risk is lower than that of breast cancer. *PALB2* mutations have been found in 2.1–4.9% of FPC kindreds [11,30,31]. According to a study of 524 families with germline *PALB2* mutations, the risk of pancreatic cancer was estimated to be 2–3% by age 80, compared to an estimated 53% risk of breast cancer in women [32].

### 2.3. STK11

Serine/threonine kinase 11 (*STK11*) regulates cell polarity, growth and proliferation and DNA damage response and function as a tumor suppressor gene. Germline *STK11* mutations are associated with Peutz–Jeghers syndrome (PJS), which gives rise to hamartomatous polyps of the gastrointestinal tract, pigmented macules on the lips and buccal mucosa, and a variety of gastrointestinal malignancies in affected individuals [33,34,35,36]. Germline *STK11* mutant carriers have an increased risk of pancreatic cancer, with 11–36% lifetime risk (by age 70), as well as other malignancies [35,37,38]. 

One study found 36 malignant tumors in 119 patients with germline *STK11* mutations. The relative risk for all cancers was 15.1 (95% CI: 10.5–21.2), with high relative risk for gastrointestinal cancers (126.2 (95% CI: 73.3–203.4)) and for pancreatic cancer in particular (139.7 (95% CI: 61.1–276.4)) [39]. Somatic *STK11* mutations have been identified in approximately 4% of sporadic pancreatic cancers [34,36,40]. Inactivation of *STK11* plays an important role in carcinogenesis in both sporadic and hereditary pancreatic cancer [34].

### 2.4. Lynch Syndrome

Lynch syndrome is caused by a germline mutation in one of several DNA mismatch repair (MMR) genes: MutL homolog 1 (*MLH1*), MutS homolog 6 (*MSH6*), and post-meiotic segregation 2 (*PMS2*) or loss of expression of MutS homolog 2 (*MSH2*) due to deletion in the *EPCAM.* Large deletions in the *EPCAM* gene causes transcriptional read-through into the neighboring *MSH2* gene, followed by epigenetic silencing [41]. Lynch syndrome is associated with an increased risk of colorectal cancer, endometrial cancer, and several other malignancies. Brisk lymphocytic infiltration, germinal center-producing lymphoid reaction, and several histological types (mucinous, signet ring cell, or medullary) are common pathological characteristics of cancers resulting from Lynch syndrome [42,43].

In a study of 147 families with germline mutation in the MMR gene, 31 families (21%) reported at least one case of pancreatic cancer. The cumulative risk of pancreatic cancer in these families was 1.3% (95% CI: 0.3–2.3%) by age 50 and 3.7% (95% CI: 1.5–5.9%) by age 70. The relative risk for pancreatic cancer in these families was 8.6 (95% CI: 4.7–15.7) when compared with the general population [44]. Cancer risk in Lynch syndrome differs among causative susceptibility genes. A recent prospective observational study of 3119 MMR-mutation carriers found the cumulative incidence for colon cancer, endometrial cancer, and pancreatic cancer in germline *MLH1* mutation to be 25.7% (95% CI: 20.7–30.7%), 18.7% (95% CI: 12.9–24.5%), and 1.1% (95% CI: 0.0–2.1%) by age 50, respectively, and 41.6% (95% CI: 35.2–48.0%), 40.3% (95% CI: 31.5–49.1%), and 3.9% (95% CI: 1.4–6.4%) by age 70, respectively. Although colorectal and endometrial cancer arise at a younger age in Lynch syndrome than those arising in the general population, the overall risk for pancreatic cancer is lower and occur at higher ages than colorectal cancer or endometrial cancer. Although colorectal cancer and endometrial cancer risks are similar in *MLH1* and *MSH2* mutation carriers, overall risk for pancreatic cancer in germline mutation *MSH2* was 0.5% (95% CI: 0.0–1.5%) by age 75. *MSH2*, *MSH6,* or *PMS2* germline mutation carriers were not at higher risk for pancreatic cancer [45].

Microsatellites are regions of repetitive nucleotide sequences where DNA mismatches commonly occur. Microsatellite regions in tumors of germline DNA MMR genes mutation carriers are more expanded relative to normal tissue. Microsatellite instability (MSI) is the genetic alteration associated with tumors in Lynch syndrome. Tumors are classified as MSI-indeterminant (MSI-I) if MSI is found in one microsatellite region, as MSI-high (MSI-H) if found in two or more regions, and as microsatellite stability (MSS) if there is no MSI. In a study of 15,045 individuals with solid tumors, MSI status was evaluated and germline MMR genes (*MLH1, MSH2, MSH6, PMS2,* and *EPCAM*) immunohistochemical staining for MMR protein were analyzed. Lynch syndrome was identified in 103 patients. The prevalence of Lynch syndrome in patients with MSI-H was 16% (53 of 326). MSI-H and MSI-I were observed in 51% and 13% of the Lynch syndrome patients, respectively [46]. Thus, MSI is not specific for Lynch syndrome. Somatic promoter hypermethylation of MLH1 causes loss of MLH1 function and leads to MSI [42,47]. Biallelic somatic mutation and inactivation of one of the MMR proteins are two other causes of MSI [48,49,50].

### 2.5. Familial Atypical Multiple Mole and Melanoma (FAMMM) Syndrome 

Familial atypical multiple mole and melanoma (FAMMM) syndrome is associated with multiple nevi, cutaneous, and ocular malignant melanomas, as well as pancreatic, breast, lung, and endometrium cancer. Mutations in the cyclin-dependent kinase inhibitor 2A (*CDKN2A*) gene (*p16)* cause FAMMM syndrome. The *p16* protein is a negative regulator of cell cycle progression at the G1/S checkpoint [51,52].

A specific 19-base-pair deletion in the *p16* gene (the p16 Leiden mutation) has been identified as being responsible for the FAMMM variant associated with pancreatic carcinoma. The cumulative risk of pancreatic cancer is 17% (95% CI: 3–30%) by age 75 [53]. The relative risk for pancreatic cancer in FAMMM is 13.1 (95% CI: 1.5–47.4) [54]. One study found 279 cancers in 22 families with germline *p16* Leiden founder mutation, of which 122 (44%) were melanoma, 25 (21%) were non-melanoma skin cancer, 22 (15%) were pancreatic cancer, 16 (11%) were lung cancer, and 12 (9%) were breast cancer. Relative risk for pancreatic cancer in this cohort was 46.6 (95% CI: 24.7–76.4), suggesting a high risk of pancreatic cancer in *p16* mutation carriers [55].

### 2.6. ATM

Ataxia-telangiectasia (AT) is an autosomal recessive disorder characterized by progressive cerebellar degeneration, oculocutaneous telangiectasia, immunodeficiency, and susceptibility to cancer. Pathogenic variants in the ataxia telangiectasia mutated gene (*ATM*) are the cause of AT. ATM kinase is involved in the detection of DNA damage and halts the cell cycle in the presence of DNA damage [56]. ATM kinase is involved in the phosphorylation of several key proteins including *TP53* and *BRCA1* proteins [57,58]. 

AT is generally categorized into three phenotypes (classic AT, variant AT, and AT heterozygous). AT heterozygous (individuals with a single pathogenic *ATM* variant) have none of the classic clinical manifestations of AT, but have increased cancer risk. In a study of 4607 *ATM* pathogenic variant carriers and 623,135 controls, variant carriers reported a personal history of ductal invasive breast cancer (33.4%), prostate cancer (32.8%), ovarian cancer (4.4%), colorectal cancer (2.9%), and pancreatic cancer (1.4%). The relative risk for all *ATM* pathogenic variant carriers was 2.03 (95% CI: 1.89–2.19) for ductal invasive breast cancer, 2.58 (95% CI: 1.93–3.44) for prostate cancer, 1.57 (95% CI: 1.35–1.83) for ovarian cancer, 1.49 (95% CI: 1.24–1.79) for colorectal cancer, and 4.21 (95% CI: 3.24–5.47) for pancreatic cancer [59]. Although the lifetime risk for pancreatic cancer remains uncertain, the *ATM* pathogenic variant is associated with an increased risk of pancreatic cancer [60,61].

### 2.7. Li–Fraumeni Syndrome

Li–Fraumeni syndrome is an inherited autosomal dominant disorder caused by the *TP53* gene, which is associated with a wide range of malignancies arising at an early age [62]. Risk of premenopausal breast cancer is markedly increased in women with Li–Fraumeni syndrome. The cumulative incidence rate for breast cancer in women, soft tissue sarcoma, brain tumors, and osteosarcoma by the age of 70 were reported to be 54%, 15%, 6%, and 5%, respectively [63]. In the international agency for research on cancer (IARC) germline *TP53* database, pancreatic cancer occurred in 1.2% of the affected individuals, with a relatively young median age of 53 years at diagnosis [64]. 

### 2.8. Familial Adenomatous Polyposis (FAP)

FAP is an autosomal dominant disease caused by germline pathogenic variants in the tumor suppressor gene *APC*. It is characterized by the presence of hundreds of adenomatous colorectal polyps that have the potential to become adenocarcinomas. Mutations of both alleles of *APC* lead to the absence of functional APC protein and the accumulation of beta-catenin, as well as transcriptional activation of the *Wnt*-signaling pathway [65]. Somatic *APC* mutations are also found in up to 80% of sporadic colorectal adenomas and adenocarcinomas, playing a key role in carcinogenesis.

Patients with familial adenomatous polyposis (FAP) are also at risk for extraintestinal tumors. In a study of 1391 affected individuals, FAP patients had relative risk of 4.5 (95% CI: 1.2–11.4) and cumulative lifetime risk (by age 80) of 1.7% for pancreatic cancer [66].

## 3. Hereditary Pancreatitis

Hereditary pancreatitis refers to acute recurrent or chronic pancreatitis caused by autosomal dominant variants in the PRSS1 gene (serine protease 1), SPINK1 (serine protease inhibitor kazal type 1), or other genes [67,68,69].

Chronic inflammation in hereditary pancreatitis accelerates genetic mutation and promotes the development of pancreatic cancer [70]. Hereditary pancreatitis is associated with increased risk of pancreatic cancer; although, it accounts for a very small fraction of all cases [70,71,72,73].

According SEER (National Cancer Institute’s Surveillance, Epidemiology, and End Results) registry study on 112 affected families in Europe, the lifetime risk of pancreatic cancer is 7.2–18.8%, and the relative risk was 59–67 [71,74]. The risk of pancreatic cancer increased after age 50 in both studies, with markedly increased risk in smokers and diabetics [75].

## 4. Familial Pancreatic Cancer

Familial pancreatic cancer is defined as pancreatic cancer arising in a patient with at least two FDRs with pancreatic cancer in the absence of an identified genetic cause.

A prospective registry-based study of 5179 individuals from 838 families evaluated standardized incidence ratios of pancreatic cancer based on SEER data [9]. The risk for pancreatic cancer increased in line with the number of affected FDRs. The relative risk for pancreatic cancer was 4.5, 6.4, and 32.0 in individuals with one, two, and three or more affected FDRs, respectively.

A meta-analysis of family history and pancreatic cancer from nine studies showed that the relative risk for pancreatic cancer was 1.80 in individuals with a family history of pancreatic cancer, regardless of degree of relatedness [76]. A pooled analysis conducted by the Pancreatic Cancer Cohort Consortium including Mayo Clinic Molecular Epidemiology of Pancreatic Cancer Study (case-control study) [77] and ten cohort studies [78,79,80,81,82,83,84,85,86,87] reported relative risk of 1.7 and 4.26 in individuals with one and two or more affected FDRs, respectively [88]. These analyses confirm the relationship between strong family history and risk for pancreatic cancer.

Previous studies found susceptibility gene mutations in about 20% of pancreatic cancer patients with strong family history. On the flipside, deleterious germline mutations were not detected in 80% of the cases. Further investigations are needed to identify susceptibility genes of familial pancreatic cancer.

## 5. Carcinogenesis of Hereditary Pancreatic Cancer

Germline mutations in *BRCA1, BRCA2, CDKN2A,* mismatch repair genes (*MLH1, MSH2, MSH6, PMS2,* and *EPCAM*), *ATM, PALB2, STK11,* and *TP53* are commonly found in hereditary pancreatic cancer. In general, inherited pancreatic cancer susceptibility syndromes have a germline mutation in one allele of these susceptible genes. Although the dominant negative effect can explain the loss of function of *TP53* despite a mutation in only one allele, carcinogenesis resulting from tumor suppressor genes requires the inactivation of second allele in most cases [89]. The second allele is inactivated by somatic mutation, loss of heterozygosity (LOH), or epigenetic silencing due to promoter hypermethylation.

### 5.1. Murine Models of Hereditary Pancreatic Cancer

Multiple combinations of genetic mutations are commonly identified in pancreatic cancer [6,90,91,92]. Germline mutations in *BRCA1/2* are most commonly associated with hereditary pancreatic cancer [17,93]. The role of *BRCA2* in FPC was evaluated using murine models of pancreatic cancer associated with *Brca2* inactivation. *Pdx1*-Cre mediated murine models demonstrated that a cooperative effect between *Brca2* inactivation and other genetic change such as *Trp53* inactivation is necessary to promote pancreatic carcinogenesis in murine models [13,14].

Inactivation of *Brca2* does not promote murine pancreatic cancer formation on its own. No mice with wild type *Brca2* or only inactivation of one or both *Brca2* alleles developed pancreatic intraepithelial neoplasia precursor lesions (PanIN) within 24 months.

Inactivation of *Brca2* significantly promotes pancreatic cancer development when combined with *Trp53* disruption, as evidenced by pancreatic cancer observed in *Trp53* null mice. Homozygous *Brca2* inactivation caused more pancreatic tumors than heterozygous *Brca2* inactivation. Although *Trp53* disruption mainly led to the development of acinar and undifferentiated pancreatic tumors, those histologically similar to human pancreatic ductal adenocarcinomas (PDAC) were observed in the *Trp53* null mice with homozygous and heterozygous Brca2 mutations. High-grade, CK19-negative undifferentiated type carcinomas were more common in homozygous *Brca2* mutations than in heterozygous mutations in these mice.

*KRAS* is mutated in 90% of human pancreatic cancers [6]. Oncogenic *KRAS* activation in pancreatic ducts lead to in PanIN, which in turn develop into PDAC [94]. Evaluation of a *Pdx1*-Cre-mediated *Kras^G12D^* mouse model revealed no significant difference in PanIN occurrence rates between wild type *Brca2* and single *Brca2* allele inactivation in such mice. Interestingly, inactivation of both alleles of *Brca2* gave rise to less pancreatic tumors and PanIN lesions than wild type *Brca2* but caused more pancreatic insufficiency due to inflammatory degeneration of pancreatic parenchyma into adipose tissue. The loss of *Brca2* in cells expressing mutant *Kras* with wild type *Trp53* lead to cell death because cell cycle checkpoints for DNA damage remain intact. Tumors in these *Brca2* disrupted mice with intact *Trp53* and activated mutant *Kras* were positive for Trp53 immunohistochemical staining and showed missense mutation of *Trp53*. Inactivation of *Trp53* appears to be necessary for *Kras*-mutated tumors to occur in these murine models with homozygous *Brca2* mutation. Previous studies reported similar prevalence of *KRAS* mutation in hereditary pancreatic cancer and sporadic pancreatic cancer [15,95,96,97]. In summary, carcinogenesis appears to result from *KRAS* activation and inactivation of *BRCA2* in combination with other genetic mutations.

Mutant *Brca2* promotes carcinogenesis in mice with both of mutant *Trp53* and mutant *Kras^G12D^*. Unlike mice with wild-type *Trp53* and mutant *Kras^G12D^*, more PDAC and decreased PDAC-free survival was observed in mice with homozygous *Brca2* inactivation than with heterozygous *Brca2* mutation or wild type *Brca2*. There was also a statistically significant reduction in PDAC-free survival in heterozygous *Brca2* mutation compared to wild type *Brca2*. Homozygous and heterozygous *Brca2* mutation with mutant *Trp53* and mutant *Kras^G12D^* caused metastatic adenocarcinoma with desmoplastic reaction evolving from PanIN. Acinar carcinomas and neuroendocrine carcinomas only developed in mice with homozygous *Brca2* mutation. The combination of *Brca2* inactivation and disrupted *Trp53* accelerates the evolution of oncogenic *Kras*-driven pancreatic cancers. Homozygous *Brca2* mutation in cells with intact checkpoint gene *Trp53* may favor apoptosis. In the setting of inactive *Trp53*, inactivation of both Brca2 alleles leads to rapid tumor progression. Inactivation of *Brca2* occurred simultaneously in these *Pdx1*-Cre mediated murine models. In human carcinogenesis, the inactivation of the second *BRCA2* allele may occur subsequent to the inactivation of checkpoint genes such as *TP53*; LOH of *BRCA2* appears to be a late event. Simultaneous loss of both *Brca2* alleles may trigger carcinogenesis of variant histopathology, such as acinar cell carcinoma. Early or late inactivation of the second allele of the wild-type *BRCA2* may affect phenotypes of human pancreatic tumors.

### 5.2. Loss of Heterozygosity

Complete loss of *BRCA2* function caused by LOH is associated with sensitivity to DNA damaging agents such as platinum agents or PARP inhibitors. Cells with homozygous *BRCA* mutation are significantly more sensitive to platinum agents and PARP inhibitors than cells with heterozygous mutations in vitro and in murine models of pancreatic, breast, and ovarian cancers [13,14,98,99,100]. 

Whole genome sequencing from several tumors has demonstrated mutational patterns associated with HRR deficiency [101]. A distinct pattern of base-substitution mutations known as signature 3 is associated with biallelic *BRCA1/2* inactivation and HRR deficiency [102,103]. An increase in signature 3 mutations was not observed in tumors with single allele inactivation of the *BRCA* gene in a study on breast cancer patients. Signature 3 was associated with tumors with biallelic inactivation of *BRCA1/2*. Tumors with biallelic *BRCA1/2* inactivation were associated with extensive LOH. LOH of chromosomes 17 and 13 was associated with inactivation of *TP53* and *RB1*. In contrast, tumors with inactivation of single allele of *BRCA1/2* could not be distinguished from those without germline variants. Thus, the allelic status of germline variants plays a role in determining tumor phenotype [104].

Genomic instability score calculated based on LOH, large-scale state transitions, and telomeric allelic imbalances was associated with sensitivity to platinum agents in a study of whole genome sequencing data of pancreatic cancer patients with pathogenic variants of *BRCA1/2* or *PALB2* [15]. Further studies are desirable to elucidate the genetic and clinical impact of allelic status of germline variants in pancreatic cancer.

### 5.3. Intraductal Papillary Mucinous Neoplasm (IPMN) and Hereditary and Familial Pancreatic Cancer

Pancreatic ductal adenocarcinomas represent the most common type of pancreatic cancer and arise from pancreatic intraepithelial neoplasms (PanINs) [94]. Intraductal papillary mucinous neoplasm of the pancreas (IPMN) is one of the cystic neoplasms of the pancreas. IPMNs progress from a benign neoplasm to invasive carcinoma of the pancreas (IPMN-derived carcinoma) in some patients over time [105]. In a study of 1404 consecutive patients with branch-duct IPMNs, 68 patients with pancreatic cancer (38 patients with IPMN-derived carcinomas and 30 patients with concomitant pancreatic ductal adenocarcinomas) were identified. The overall incidence of pancreatic cancer 5, 10, and 15 years after IPMN diagnosis were 3.3, 6.6, and 15 percent, respectively [106]. Although the true incidence of IPMN is not known, a series of 616 consecutive magnetic resonance imagings performed in unselected adults without history of pancreatic diseases found pancreatic cysts in 83 (13.5%) and many of these were likely IPMNs [107]. IPMN may be more common in patients with a family history of pancreatic cancer or hereditary pancreatic cancer syndromes [108,109,110,111]. 

IPMNs are often multifocal, and this multifocality suggests a possible underlying genetic predisposition to IPMNs. It is not clear whether pancreatic cancer susceptibility genes such as *BRCA1, BRCA2, CDKN2A,* mismatch repair genes (*MLH1, MSH2, MSH6, PMS2,* and *EPCAM*), *ATM, PALB2, STK11,* and *TP53* are associated with IPMNs. In a study of 350 patients with surgically resected IPMNs, 26 germline mutations in 23 patients (7.3%) including 10 germline mutations (established pancreatic cancer susceptibility genes) in 9 patients (2.9%) [112]. There was no significant difference in prevalence of germline mutations in these genes between IPMN patients and unselected pancreatic ductal adenocarcinoma patients. However, germline mutations in *ATM* were significantly more common in patients with an IPMN than in controls. The patients with IPMN with these pathogenic germline mutations were more likely to have concomitant pancreatic ductal adenocarcinomas than IPMN patients without these mutations. Analysis of whole exome or targeted sequence in 148 samples from IPMNs and mucinous cystic neoplasms identified alterations in *ATM* in 17% of the cases [113]. Genetic analysis of IPMNs highlighted the link between mutations in *ATM* and IPMNs [112,113].

## 6. Prevalence of Germline Mutations in Pancreatic Cancer

Prevalence of germline mutations in sporadic pancreatic cancer is summarized in Table 2. In the largest study of 3030 patients with pancreatic cancer, pathogenic germline mutations were identified in 249 patients (8.2%). Pathogenic germline variants of *BRCA1, BRCA2, PALB2,* and *ATM* were found in 0.59%, 1.95%, 0.4%, and 2.28% of cases, respectively [114]. Lynch syndrome, defined by mutations causing inactivation of MMR genes, was found in 0.5% of cases (*MLH1*: 0.17%; *MSH2*: 0.03%; *MSH6*: 0.23%; and *PMS2*: 0.07%). Pathogenic variants in *CDKN2A* and *TP53* were identified in 0.33% and 0.20% of cases, respectively.

In a study on 1005 Japanese pancreatic cancer patients, 6.67% had pathogenic germline mutations. Pathogenic variants were identified in *BRCA1* (0.90%), *BRCA2* (2.49%), *PALB2* (0.20%), and *ATM* (1.69%). Lynch syndrome was identified in 0.5% of cases (*MLH1*: 0.10%; *MSH2*: 0%; *MSH6*: 0.30%; *PMS2*: 0%; and *EPCAM*: 0.10%). Li–Fraumeni syndrome and FAMMM were only identified 0.2% and 0.1% of cases, respectively [115].

Germline mutations in pancreatic cancer susceptibility genes are commonly identified in pancreatic cancer patients without a significant family history of cancer [116,117]. In a study of 854 pancreatic cancer patients, 33 patients (3.9%) had pathogenic germline mutations, of which only 9% had a family history of pancreatic cancer. On the other hand, 14.3% of patients without germline mutations had a family history of pancreatic cancer, of which 73.5% had an FDR with pancreatic cancer [116].

A causative germline mutation is identified in 20% of pancreatic cancer patients with a strong family history or a personal history of malignancy [118,119]. In a study of comparing prevalence of deleterious germline mutations (*BRCA1/2*, *PALB2*, and *CDKN2A*) in FPC (having two FDRs with pancreatic cancer) and non-FPC (at least two affected relatives, but no FDRs) families, FPC families have more germline mutations (8.0%) than non-FPC families (3.5%) (odds ratio: 2.4; 95% CI: 1.06–5.44; *p* = 0.03) [7]. The prevalence of the four deleterious germline mutations in FPC was as follows: *BRCA1*: 1.2%; *BRCA2*: 3.7%; *PALB2*: 0.6%; and *CDKN2A*: 2.5%. In a study of 303 pancreatic cancer patients in the Mayo Clinic Familial Pancreatic Cancer Registry, 25 cancer susceptibility genes were sequenced using a hereditary cancer panel. In the study, 12.9% of 186 patients who had two FDRs with pancreatic cancer and 9.4% of 117 patients not meeting the criteria for FPC were carriers [120]. In a study of 81 Japanese FPC patients, possibly deleterious germline variants were detected in 18 (22.2% of cases): *ATM*: 7; *BRCA2*: 3; *ASXL1*: 2; *DNMT3A*: 2; *ERCC4*: 2; and *MLH1*: 2 [121].

Whole-exome sequencing has revealed new possibly deleterious germline variants in *FAT* family members [121]. Germline susceptibility gene mutations were not identified in 80% of pancreatic cancer patients with strong family history. Further studies are needed to identify more candidate genes associated with FPC.

## 7. Surveillance Strategy for High-Risk Cases

The optimal screening strategy for detecting early pancreatic cancer in the general population remains unclear. The United States Preventive Services Task Force (UPSTF) recommends against population-based screening for average-risk patients [122]. Individuals with a fivefold increase in pancreatic cancer risk or a lifetime risk of over 5% are believed to be candidates for screening. Some germline susceptibility gene mutations do not lead to sharp increases in pancreatic cancer risk on their own. In addition, non-familial risk factors such as smoking, alcohol consumption, diet, high fasting plasma glucose, and high body mass index have been associated with higher risk of pancreatic cancer [118,123]. Several groups have published guidelines for screening for high-risk patients [12,16,124].

The International Cancer of the Pancreas Screening (CAPS) Consortium summit in April 2018 identified potential candidates for screening, as follows: (1) individuals with germline mutations in *LKB/STK11* (Peutz–Jeghers syndrome) and *CDKN2A* (FAMMM) regardless of family history; (2) individuals with germline mutations in *BRCA2, PALB2, ATM,* and *MLH1*/*MSH2/MSH6* (Lynch syndrome) with at least one affected FDR; and (3) individuals with at least two affected relatives on the same side of the family, of whom at least one is an FDR regardless of gene mutation status. Note that screening is not recommended in individuals with germline mutations in *BRCA1* with one affected FDR [12]. On the other hand, the American Gastroenterological Association (AGA) Institute Clinical Practice recommends screening (1) individuals with germline mutations in *LKB/STK11* (Peutz–Jeghers syndrome), *CDKN2A* (FAMMM), and *PRSS1* (hereditary pancreatitis) regardless of family history; (2) individuals with germline mutations in *BRCA1, BRCA2, PALB2, ATM,* or *MLH1/MSH2/MSH6* (Lynch syndrome) and at least one affected FDR; and (3) FDRs of patients with pancreatic cancer with at least two affected genetically related relatives [16]. Recommendations from the two groups differ with respect to patients with pathogenic *BRCA1* variants, hereditary pancreatic cancer, and family history. The risk for pancreatic cancer in individuals with pathogenic variants of *BRCA1, BRCA2, PALB2,* or *ATM* but without a history of pancreatic cancer has not been elucidated. In a study of 204 individuals with pathogenic *BRCA1/2* variants, all *BRCA1/2* patients irrespective of pancreatic cancer family history were recommended to undergo surveillance because the prevalence of pancreatic abnormalities was not affected by family history [125].

Screening for FPC relatives (with no known germline mutations) should begin at age 50 or 10 years younger than the youngest relative with pancreatic cancer. For germline mutation carriers, screening should begin at age 45 or 50. Screening should be initiated at age 40 in *CDKN2A* mutation carriers and at age 30–40 for patients with Peutz–Jegher syndrome [12,16]. Although severe chronic pancreatitis makes surveillance challenging, screening for patients with hereditary pancreatitis should begin at age 40 [16].

A cross-sectional blinded comparison of endoscopic ultrasound (EUS), magnetic resonance imaging (MRI), and computed tomography (CT) demonstrated that EUS and MRI were comparable and better than CT in detecting pancreatic lesions in high-risk patients [126]. MRI in combination with EUS is recommended as the screening modalities. For patients with low-risk findings by a multidisciplinary team, the recommended surveillance interval is 12 months.

A cohort study screened 354 individuals at high risk for pancreatic cancer by EUS, MRI and/or CT [127]. After a median follow-up of 5.6 years, 24 (7%) developed neoplasms. Of the 14 PDACs identified, 10 (71%) were detected by surveillance and nine of these were resectable. By contrast, four cases were diagnosed outside of the surveillance program, all of which were symptomatic and unresectable. Three-year OS was higher when cancer was discovered within the program than outside the program (85% versus 25%). In an MRI screening study on 79 *CDKN2A* mutation carriers, pancreatic cancer was diagnosed in 7 patients (9%) during a median follow-up period of 4 years [128]. All cases were resectable at diagnosis. While these studies illustrated the benefits of surveillance, large prospective studies are necessary to demonstrate the clinical benefit of pancreatic cancer screening in high-risk individuals.

## 8. Medication for Patients with Susceptibility Gene Mutations

### 8.1. Platinum Agents

Mutations in *BRCA1*, *BRCA2*, and *PALB2* are associated with defective homologous recombination after DNA damage. This deficiency in the repair of DNA double-strand breaks increases vulnerability to agents that damage DNA [129].

In breast and ovarian cancers, sensitivity to platinum agents in germline HRR deficiency patients has been shown [130]. In a retrospective study of 71 *BRCA1/2* associated pancreatic cancer, unresectable pancreatic cancer patients treated with platinum agents had significantly longer OS than those treated with non-platinum agents (22 vs. 9 months; *p* = 0.039) [131]. Despite being based on a very low number of evaluable patients, a meta-analysis of six studies comparing platinum agents to non-platinum agents in germline BRCA mutant unresectable pancreatic cancer patients reported significantly longer OS in the platinum group than in the non-platinum group (23.7 vs. 12.2 months; mean difference of 10.2 months, 95% CI 5.07–15.37; *p* < 0.001) [132]. From a study of 262 patients who underwent both germline and somatic MSK-IMPACT analysis, patients with homologous recombination deficiency had significantly improved median progression-free survival (PFS) after first-line treatment with platinum agents rather than non-platinum agents (12.6 (95% CI, 9.6–24.9) vs. 4.4 (95% CI, 3.0–10.0) months). Core homologous recombination mutations such as *BRCA1/2* and *PALB2* and biallelic homologous recombination mutations were associated with genomic instability and sensitivity to first-line platinum agents [97]. In pancreatic cancer mouse models, cells with heterozygous *Brca* mutations are significantly less sensitive to platinum agents than those with homozygous mutations [13,14]. In a study of whole genome sequencing data of pancreatic cancer patients with pathogenic variants of *BRCA1/2* or *PALB2*, signature 3 mutations, and genomic instability were correlated with platinum response [15]. 

Although there is no clear evidence for superiority of first-line platinum agents over non-platinum agents in pancreatic cancer for *BRCA1/2* or *PALB2* mutant carriers, the latest National Comprehensive Cancer Network guidelines recommend FOLFIRINOX or modified FOLFIRINOX or gemcitabine plus cisplatin as first-line chemotherapy for known *BRCA1/2* or *PALB2* mutated cases [133].

### 8.2. PARP Inhibitors

DNA damages can be classified into single-strand breaks (SSB) and double-strand breaks (DSB). MMR pathways, nucleotide excision repair (NER) pathways, and base excision repair (BER) pathways play an important role in SSB. Poly (ADP-ribose) polymerase proteins 1 and 2 (PARP1 and 2) are recruited to the SSB site and involved in BER. Tumor suppressor proteins BRCA1 and BRCA2 play a crucial role in the mechanism of DSB repairs. BRCA2 interacting with PALB2 and RAD51 are recruited to the DSB site and move the damaged DNA strand close to the sister chromatid, to be used as a template for the HRR process. A dysfunction of one DNA damages repair (DDR) in tumors make tumor cells dependent on the remaining undamaged DDR pathways. PARP inhibition prevents tumor cells from repairing their SSBs and causes DSBs. In germline mutant *BRCA* tumors, DSB leads to irreversible cellular damage and apoptosis [134,135].

In the phase III POLO trial, 154 platinum-responsive metastatic pancreatic cancer patients were randomly assigned to receive the PARP inhibitor olaparib or placebo as maintenance therapy. PFS was significantly longer in the olaparib group (7.4 vs. 3.8 months, *p* = 0.004) and objective response rate (ORR) was significantly higher in the olaparib group (23% vs. 12%) [136]. A phase II study evaluated maintenance rucaparib in patients with platinum-responsive pancreatic cancer and a pathogenic germline and somatic variant in *BRCA1*, *BRCA2*, or *PALB2*. Results were promising, with a 37% ORR in somatic or germline mutant *BRCA1/2/PALB2* patients [137].

PARP inhibitor monotherapy was also evaluated. Olaparib and rucaparib were tested in phase II trials including approximately 20 germline BRCA1/2 mutant pancreatic cancer patients. ORRs were 21.1% and 21.7%, respectively [138,139]. Veliparib monotherapy showed disappointing results with a 0% ORR and median PFS and OS of 1.7 and 3.1 months, respectively. PARP-trapping activity of veliparib was lower than those of olaparib and rucaparib [140]. Telazoparib, a selective last generation PARP inhibitor with the strongest PARP-trapping activity, was also evaluated in a phase I study [141].

The use of PARP inhibitors in combination with chemotherapy has also been studied. Although veliparib has the lowest trapping activity among PARP inhibitors, it was evaluated in first-line treatment in combination with chemotherapy. In a phase I trial of gemcitabine plus cisplatin plus veliparib, ORR was 77.8% and median OS was 23.3 months in germline mutant BRCA cases. A randomized phase II trial of this regimen is currently ongoing [142].

Tumor mutational burden and inflammatory activity are associated with DDR deficiency. Following promising results from breast and ovarian cancer, PARP inhibitors in combination with immune checkpoint inhibitors are being investigated for germline mutant *BRCA1/2* pancreatic cancer [143]. 

Outcomes of treatment with PARP inhibitors in pancreatic cancer patients with BRCAness (*ATM, BAP1, BARD1, BLM, BRIP1, CHEK2, FAM175A, FANCA, FANCC, NBN, PALB2, RAD50, RAD51, RAD51C,* and *RTEL1*), pathogenic variants in genes associated with HRR deficiency, remain unclear [144]. Further studies are desirable for this population.

### 8.3. Immune Checkpoint Inhibitors

The US Food and Drug Administration approved anti PD-1 therapy with pembrolizumab for solid tumors with high levels of microsatellite instability (MSI-H) or deficient mismatch repair (dMMR) [145]. DNA MMR genes found in Lynch syndrome such as *MLH1, MSH2, MSH6,* and *PMS2* play a role in repairing DNA damage resulting from single base pair insertions or deletions. Mutations accumulate due to an inability to repair DNA damage. Almost all tumors with dMMR demonstrate a high tumor mutation burden. MSI-H reflects underlying deficiency in MMR capability in regions with short, repetitive DNA sequences [46,146]. Mutations in the tumor genome can cause tumors to express mutant proteins, some of which are tumor-specific, mutation-derived antigens (neoantigens), which can be recognized and targeted by the immune system [147,148].

In a study of 86 patients with dMMR across 12 different tumor types, ORR was 53.4% and the complete response (CR) rate was 20.9%. ORR of 62% and a CR rate of 25% were achieved in the eight pancreatic cancer patients in this study [149].

## 9. Conclusions and Future Perspectives

Although several causative genes of hereditary pancreatic cancers have been identified, further investigations are necessary to identify susceptibility genes in FPC. Clinical features and genomic backgrounds are heterogeneous. It must also be noted that carcinogenesis in pancreatic cancer may arise independently of inherited germline mutations in a subset of pathogenic germline variant carriers.

Although several guidelines for screening have been published, surveillance standards of pancreatic cancer for high-risk cases have not been established. PancPRO, the first risk prediction model for pancreatic cancer, is no longer available online for clinical use [150]. A statistical model using artificial intelligence should be developed to assess the risk of pancreatic cancer based on personal history, family history, and inherited germline pathogenic mutations of susceptibility genes for pancreatic cancer.

With respect to medications, maintenance treatment with the PARP inhibitor olaparib is currently available for platinum-responsive *BRCA*-mutated pancreatic cancer. Randomized trials on PARP inhibitors in combination with cytotoxic agents are in progress. Biomarkers such as signature 3 based on whole genome sequencing may identify patients other than those with pathogenic germline *BRCA* variants who stand to benefit from such treatment.

Genomic and clinical data on hereditary pancreatic cancer syndromes and FPC are still limited. Worldwide spread of the FPC registration system and large-scale research studies may lead to further insight, early detection of pancreatic cancer, and improved treatment in affected individuals going forward.

## Figures and Tables

**Table 1 ijms-23-01205-t001:** Causative genes and clinical features of hereditary pancreatic cancer syndromes.

Syndrome	Causative Gene	Relative Risk for Pancreatic Cancer	Lifetime Risk for Pancreatic Cancer
HBOC	*BRCA1*	2.3	1%
HBOC	*BRCA2*	3.5–10	4.9%
	*PALB2*	2.4	2–3%
	*ATM*	4.2	
Lynch syndrome	*MLH1, MSH2, MSH6, PMS2, EpCAM*	8.6	3.7%
FAMMM	*CDKN2A*	13.1–22	17%
Peutz–Jeghers syndrome	*STK11*	139.7	11–36%
Li–Fraumeni syndrome	*TP53*	7.3	-
FAP	*APC*	4.5	1.7%
Hereditary pancreatitis	*PRSS1, SPINK1*	59–67	7.2–18.8%
Familial pancreatic cancer number of FDR (one, two, three or more)	unknown	4.5, 6.4, 32.0	-
Family history (one affected FDR) (unselected population)		1.7	

Abbreviations: HBOC, hereditary breast and ovarian cancer syndrome; FAMM, familial atypical multiple mole melanoma; FAP, familial adenomatous polyposis; FDR, first degree relative.

**Table 2 ijms-23-01205-t002:** Germline mutation prevalence in pancreatic cancer.

	Sporadic Pancreatic Cancer (Unselected)	Pancreatic Cancer with Strong Family History
germline mutation	6.7–12.9%	(∼20% to 25%)
*BRCA1*	0.6–0.9%	1.2%
*BRCA2*	2.0–2.5%	3.7–5.6%
*PALB2*	0.2–0.4%	3.7%
*ATM*	1.7–2.3%	2.6–3.7%
*MLH1*	0.1–0.2%	1.0–1.9% (MMR genes)
*MSH2*	0.03%	-
*MSH6*	0.2–0.3%	-
*PMS2*	0.1%	-
*STK11*	<1%	-
*CDKN2A*	0.1–0.3%	2.5%
*TP53*	0.2%	-

## Data Availability

Not applicable.

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
