# Peer review of "Molecular Features and Clinical Management of Hereditary Pancreatic Cancer Syndromes and Familial Pancreatic Cancer"

_ijms, 2022, doi:10.3390/ijms23031205_

Round 1

Reviewer 1 Report

This is an excellent review. I have a key comment that would need to be addressed. The entire review addresses the risk of pancreatic cancer. Pancreatic cancer (adenocarcinoma/ PDAC) is from 2 sources - PanIN and IPMNs. This review likely focuses on PanIN lesions (so-called solid pancreatic cancer / PDAC) cancers. However, adenocarcinoma also arises from IPMNs. To be comprehensive, the authors should include one or two paragraphs on what is known about IPMNs and their risk in hereditary and familial pancreatic cancer. Notably, IPMNs are extremely common in the elderly, where studies with MRI/MRCPs (of the abdomen) have found a prevalence rate of cysts (30-45%). Most cysts are incidentally detected. With increasing prevalence of cystic lesions, the inclusion of this pancreatic pathology would be appropriate for this review. 

Author Response

We appreciate your positive comments and helpful suggestions for improving our manuscript.

Reviewer 1 comment: The authors should include one or two paragraphs on what is known about IPMNs and their risk in hereditary and familial pancreatic cancer.

Our response: We have added paragraphs titled “IPMN and hereditary and familial pancreatic cancer”. Added paragraphs are detailed below.

“5.3 Intraductal papillary mucinous neoplasm (IPMN) and hereditary and familial pancreatic cancer

Pancreatic ductal adenocarcinomas represent the most common type of pancreatic cancer and arise from pancreatic intraepithelial neoplasms (PanINs)[94]. Intraductal papillary mucinous neoplasm of the pancreas (IPMN) is one of the cystic neoplasms of the pancreas. IPMNs progress from a benign neoplasm to invasive carcinoma of the pancreas (IPMN-derived carcinoma) in some patients over time[105]. In a study of 1404 consecutive patients with branch-duct IPMNs, 68 patients with pancreatic cancer (38 patients with IPMN-derived carcinomas and 30 patients with concomitant pancreatic ductal adenocarcinomas) were identified. The overall incidence of pancreatic cancer 5, 10 and 15-years after IPMN diagnosis were 3.3, 6.6 and 15 percent respectively[106]. Although the true incidence of IPMN is not known, a series of 616 consecutive magnetic resonance imagings performed in unselected adults without history of pancreatic diseases found pancreatic cysts in 83 (13.5%) and many of these were likely IPMNs[107]. IPMN may be more common in patients with a family history of pancreatic cancer or hereditary pancreatic cancer syndromes[108-111]. 

IPMNs are often multifocal and this multifocality suggest a possible underlying genetic predisposition to IPMNs. It is not clear whether pancreatic cancer susceptibility genes such as BRCA1, BRCA2, CDKN2A, mismatch repair genes (MLH1, MSH2, MSH6, PMS2, EPCAM), ATM, PALB2, STK11, and TP53 are associated with IPMNs. In a study of 350 patients with surgically resected IPMNs, 26 germline mutations in 23 patients (7.3%) including 10 germline mutations (established pancreatic cancer susceptibility genes) in 9 patients (2.9%)[112]. There was no significant difference in prevalence of germline mutations in these genes between IPMN patients and unselected pancreatic ductal adenocarcinoma patients. However germline mutations in ATM was significantly more common in patients with an IPMN than in controls. The patients with IPMN with these pathogenic germline mutations were more likely to have concomitant pancreatic ductal adenocarcinomas than IPMN patients without these mutations. Analysis of whole exome or targeted sequence in 148 samples from IPMNs and mucinous cystic neoplasms identified alterations in ATM in 17% of the cases[113]. Genetic analysis of IPMNs highlighted the link between mutations in ATM and IPMNs[112, 113].”

Reviewer 2 Report

The review summarizes molecular features and treatments of hereditary pancreatic cancer syndromes and surveillance procedures for unaffected high-risk cases. Authors review transgenic murine models to gain a better understanding of carcinogenesis in hereditary pancreatic cancer. Genomic and clinical data on hereditary pancreatic cancer syndromes and familial pancreatic cancer are still limited. Authors conclude that worldwide spread of the FPC registration system and large-scale research studies may lead to further insight, early detection of pancreatic cancer, and improved treatment in affected individuals going forward.

The manuscript is well-written therefore I recommend it for publication after dealing with the following minor error:

All sections of the manuscript are generally numbered erroneously as 1. Authors should make the correct consecutive numbering of sections.

Author Response

We appreciate your positive comments and helpful suggestions.

Reviewer 2 comment: All sections of the manuscript are generally numbered erroneously as 1. Authors should make the correct consecutive numbering of sections.

Our response: The numbering of the downloaded latest version of the manuscript is different from originally submitted one with correct numbering. We would like to make contact with the editorial office and inquire about the numbering.